# Interest in public involvement, engagement and participation in research among patients with asthma: A secondary analysis of the EPI-ASTHMA study

Mafalda Simões Cunha[1☸], João da Silva Santos[1☸], Ana Sá-Sousa[2], Ana Luísa Neves[2,3], Ana Alves da Silva[4], Diana Gomes[4], Janete Quelhas Santos[4], Filipa Bernardo[5], Jaime Correia de Sousa [6], João Almeida-Fonseca[2,7,8], Cristina Jácome [2]*, Epi-asthma Study group[¶]

1 Faculty of Medicine, University of Porto, Porto, Portugal, 2 RISE-Health, MEDCIDS-Department of Community Medicine, Information and Health Decision Sciences, Faculty of Medicine, University of Porto, Porto, Portugal, 3 Department of Primary Care and Public Health, Imperial College London, London, United Kingdom, 4 Centre for Health Technology and Services Research (CINTESIS), Faculty of Medicine of the University of Porto, Porto, Portugal, 5 AstraZeneca, Queluz, Lisboa, Portugal, 6 University of Minho, Life and Health Sciences Research Institute (ICVS)/3B's—PT, Government Associate Laboratory, Braga, Portugal, 7 MEDIDA, Porto, Portugal, 8 Allergy Unit, Hospital and Institute CUF, Porto, Portugal

¶ Membership of the Epi-asthma Study group is provided in the Acknowledgments.
☸ These authors contributed equally to this work
* cristinajacome.ft@gmail.com

## Abstract

Patient and Public Involvement, Engagement and Participation (PPIEP) is essential in asthma research due to the disease's complexity. However, information regarding the characteristics of patients with asthma interested in engaging in PPIEP initiatives is scarce. We aimed to identify factors associated with the interest in PPIEP among patients with asthma. This is a secondary analysis of the EPI-ASTHMA nationwide asthma prevalence study, which included adult patients with asthma recruited from 38 primary healthcare centres from mainland Portugal. Demographic and clinical characteristics as well as interest in PPIEP were collected. A multivariable logistic regression model was built to explore associations between participants' characteristics and PPIEP interest. A total of 502 participants with asthma (52 ± 15.9 years, 64.1% females) were analysed and 218 (43.4%) were interested in being involved in PPIEP. Having higher education (adjusted OR 2.29, 95%CI 1.47–3.57) and owning a smartphone (adjusted OR 4.63, 95%CI 2.51–8.55) were significantly associated with higher odds of being interested in PPIEP. Being female demonstrated a borderline significant association (adjusted OR 1.53, 95%CI 1.00–2.33). Participants experiencing higher quality of life were significantly associated with lower odds of being interested in PPIEP (adjusted OR 0.77, 95%CI 0.65–0.92). Two out of five patients with asthma were interested in participating in PPIEP initiatives. By understanding the characteristics of patients willing to participate in PPIEP, we can develop strategies

**Data availability statement:** There are ethical or legal restrictions on sharing a de-identified data set, because that the data contain potentially identifying or sensitive patient information and the ethics committee imposed them. This study was approved by the ethics committees of the Regional Health Administrations of North (CE/2022/117), Center (27/2021), Lisbon and Tagus Valley (2775/CES/2022), Alentejo (11/CE/2022) and Algarve (1/2022) and of the Local Health Units of Matosinhos (38/CES/JAS) and Alto Minho (38/2021). The original approval from the Ethical Review Boards did not include direct and free data access and therefore, data cannot be made freely available. All data supporting the findings are available from the Faculty of Medicine of University of Porto for all interested researchers who meet the criteria for access to confidential data. Please contact Cristina Jácome, the corresponding author at cjacome@med.up.pt, or the Faculty of Medicine of University of Porto at fmup@med.up.pt to request the data.

**Funding:** This study was sponsored and funded by AstraZeneca. The funder provided support in the form of salaries for authors FB, but did not have any additional role in the study design, data collection and analysis, decision to publish, or preparation of the manuscript.

**Competing interests:** Filipa Bernardo (FB) is an employee of AstraZeneca, Produtos Farmacêuticos SA. The remaining authors have no conflicts of interest to declare.

to reach a wider audience and make PPIEP more inclusive. The identified associated factors need to be further investigated.

## Introduction

Asthma is a chronic respiratory disease affecting approximately 300 million people worldwide [1]. In Portugal, asthma affects 7.1% of the population [2]. Characterized by variable symptoms, asthma often remains uncontrolled in a significant proportion of patients [3]. This variation in asthma control is influenced by several factors, including adherence to treatment, inhaler technique, comorbid conditions, and environmental factors such as exposure to allergens or smoke and respiratory tract infections [4,5]. Many patients with asthma have a limited understanding of their condition, which can lead to poor health outcomes, including exacerbations, as well as psychological challenges such as anxiety, depression [6], and feelings of worthlessness and helplessness [7]. Additionally, when compared to the general population, children and young adults with asthma experience higher rates of hospitalisation and mortality from all causes [8]. Given the complex challenges of effectively managing asthma and improving patient outcomes, it is vital that patients and the public are actively involved in healthcare and research processes.

Patient and Public Involvement, Engagement and Participation (PPIEP) in scientific research refers to an active partnership involving people from the general population and patients along with community leaders or representative entities and stakeholders in a comprehensive way [9]. PPIEP can provide researchers with a deeper understanding of what it is like to live with a particular health condition. This collaborative approach fosters stronger connections between those directly affected and the research community, ensuring that research priorities align with the real-world needs of patients [10]. A review of reviews showed that incorporating PPIEP into health research and healthcare can be a powerful tool for identifying patient needs and optimising treatment approaches [11]. PPIEP aims to improve patient outcomes and reduce healthcare costs by empowering patients to make informed decisions and promoting open discussions about disease symptoms and comorbidities [12]. Yet, previous experiences in other contexts or conditions demonstrate that the effectiveness of PPIEP depends on various factors, including the quality of the relationship between researchers, patients, and the public [13], accessibility of information and training opportunities [14]; as well as the health limitations and sociodemographic characteristics of the participants [15].

Within the respiratory field, the European Lung Foundation stands out for its work in PPIEP, involving patients with chronic respiratory diseases in both research and guideline development [16]. In Portugal, the ConectAR network has shown that including patients with chronic respiratory diseases and their caregivers as co-researchers ensures that their perspectives are considered throughout the research process, from the initial planning stages to project coordination, scientific activities, and disseminating the findings [17]. A qualitative study investigating PPIEP

in asthma, demonstrated that patients were willing to share their knowledge and experience of the disease, and that their main motivations were improvements in quality of life and a sense of social responsibility [16].

While PPIEP in research is valuable, it does not inherently guarantee inclusivity. Nonetheless, making a conscious effort toward inclusive PPIEP is crucial to diversify research participation and tackle health disparities. A survey of public contributors to National Institute of Health and Social Care Research (NIHR)-funded research revealed a lack of diversity, with participants in PPIEP initiatives being mostly women (57%), over 61 years old, white, and heterosexual [18]. A study in the United Kingdom found that among participants in PPIEP activities, the majority were women (63%) with a university education (43%) or who were employed (34%), and the most common age groups were 19–24 years old (21%) followed by those 65 years and older (20%) [19]. In Portugal, the ConectAR network involves 137 members (median age 36, range 18–72 years) [17]. However, there no other information is available about their characteristics [15]. Currently, there is a lack of data regarding the characteristics of patients with asthma who participate in PPIEP studies.

We aimed to identify factors associated with the interest in being involved in a PPIEP among patients with asthma. Understanding these factors is critical for designing equitable engagement strategies and can guide researchers and policymakers in creating inclusive PPIEP frameworks.

## Materials and methods

### Study design

This study is a secondary analysis of the EPI-ASTHMA nationwide asthma prevalence study. Further details of the primary study can be found elsewhere [2,20]. We used data collected between May 2021 and March 2024 in 38 primary healthcare centres from mainland Portugal. This study was approved by the ethics committees of the Regional Health Administrations of North (CE/2022/117), Center (27/2021), Lisbon and Tagus Valley (2775/CES/2022), Alentejo (11/CE/2022) and Algarve (1/2022), as well as by those of the Local Health Units of Matosinhos (38/CES/JAS) and Alto Minho (38/2021). All the participants provided written informed consent. This study was reported according to the Strengthening the Reporting of Observational studies in Epidemiology (STROBE) guidelines [21].

### Participants

EPI-ASTHMA study comprised 4 stages, which have been described in detail in the published articles [2,20]: phone call invitation (stage 0), phone screening interview (stage 1), diagnostic visit (stage 2) and 3-month phone follow-up (stage 3). This secondary analysis included participants who were diagnosed with current asthma or possible asthma during the diagnostic visit and who agreed to answer an additional questionnaire regarding PPIEP during Stage 2. Further details about this visit and the diagnosis process can be found in the published articles [2,20]. Patients with specific physical and/or cognitive disabilities that prevented them from cooperating with the study procedures (including lung function tests) or from understanding/answering the self-reported questionnaires were excluded.

### Data collection

This secondary analysis used data collected during Stages 1 and 2. During Stage 1, socio-demographic (i.e., age, sex, marital status, education, occupation, geographic region) and clinical (i.e., smoking status) data were collected. Participants were also asked whether they owned a smartphone and whether they ever used apps to track their health or fitness. During Stage 2, participant's body mass index (BMI) was assessed, and they self-reported their comorbidities, the number of asthma exacerbations in the last year, and the number of unscheduled specialist appointments in the last year. They also answered two patient-reported outcome measures: the Control of Allergic Rhinitis and Asthma Test (CARAT) [22] and the mini–Asthma Quality of Life Questionnaire (mini-AQLQ) [23]. CARAT is a self-reported questionnaire with a total score (CARAT-T) ranging between 0–30 points. A CARAT-T > 24 indicates good disease control [22]. The mini-AQLQ includes

15 items answered on a 7-point Likert scale (where 7 no impairment and 1 significant impairment) [23]. The score for the questionnaire is the average of the responses. At the end of the Stage 2 visit, patients answered a question regarding PPIEP interest ('We intend to form a group of people with asthma to take a more active role in research. Your participation will be occasional and will contribute to enriching research projects - from their conception to the final result - with the perspective and needs of people who deal with asthma day by day. Are you interested in collaborating with this group?').

## Data analysis

Descriptive statistics were used to characterise the sociodemographic and clinical characteristics of the participants. Absolute and relative frequencies were used to characterise the categorical variables. Means and standard deviations (SD) or medians and interquartile ranges (p25-p75) were used, according to data distribution, to characterise the numerical variables.

Simple logistic regressions were used to explore associations between participant characteristics (age, sex, marital status, education, occupation, geographic region, body mass index, smoking habits, smartphone ownership, use of health and fitness apps, number of comorbidities, exacerbations in the past year, unscheduled specialist appointments in the past year, CARAT-T, mini-AQLQ scores) and PPIEP interest. The dependent variable was interest in PPIEP (1 = yes, 0 = no). Variables found to be statistically significant (p < 0.05) in the simple logistic regressions were selected to further explore their relationship with PPIEP and to adjust for possible confounders (e.g., age) in a stepwise multivariable logistic regression model. The overall models were evaluated using goodness-of-fit tests and Nagelkerke's $R^2$, and the final model was selected based on the best combination of these results. The level of significance was set at $\alpha = 0.05$. Statistical analyses were performed using IBM SPSS Statistics V.29.0.0.0 (IBM Corporation).

## Results

### Participants' characteristics

A total of 518 participants were diagnosed with asthma in the EPI-ASTHMA study, but 16 did not answer the PPIEP question. Therefore, 502 participants (97%, mean age 52 ± 15.9 years) were included in this secondary analysis. The majority were females (322, 64.1%), 153 (30.5%) had post-secondary education, and 315 (62.7%) were employed. A total of 383 participants (76.3%) owned a smartphone and almost half (48.8%) used health and fitness apps. Table 1 shows the sociodemographic and clinical characteristics of the study participants.

### Interest in participating in PPIEP

A total of 218 (43.4%) participants with asthma expressed interest in PPIEP. In the multivariable logistic regression (Table 2), having higher education (adjusted OR 2.29, 95%CI 1.47–3.57) and owning a smartphone (adjusted OR 4.63, 95%CI 2.51–8.55) were significantly associated with higher odds of interest in PPIEP. Being female demonstrated a borderline significant association with higher odds of interest in PPIEP (adjusted OR 1.53, 95%CI 1.00–2.33). Conversely, a higher quality of life (mini-AQLQ score) was significantly associated with lower odds of interest in PPIEP involvement (adjusted OR 0.77, 95% CI 0.65–0.92). This model correctly classified 69% of cases, with an Nagelkerke's $R^2$ of 0.22 (22%).

## Discussion

This study showed that two out of five patients with asthma were interested in participating in PPIEP initiatives. Factors such as higher levels of education, smartphone ownership, and female gender were associated with increased interest in PPIEP, while experiencing better quality of life was associated with reduced interest.

The frequency of interest in PPIEP is noteworthy and aligns with findings from prior research indicating a general willingness among patients with chronic respiratory diseases to engage in PPIEP efforts [24]. Understanding PPIEP interest

**Table 1. Sociodemographic and clinical characteristics of the participants (n = 502).**

| Characteristics | Total n = 502 | Interest PPIEP n = 218 | No interest PPIEP n = 284 |
|---|---|---|---|
| **Age years, Mean (SD)** | 52 (15.9) | 46.9 (14.4) | 55.1 (16.0) |
| **Age groups[a]** | | | |
| 18–64 years | 374 (74.5) | 187 (85.8) | 187 (65.8) |
| ≥65 years | 119 (23.7) | 28 (12.8) | 91 (32.0) |
| **Female** | 322 (64.1) | 152 (69.7) | 170 (59.9) |
| **Marital Status** | | | |
| Married | 284 (56.6) | 118 (54.1) | 166 (58.5) |
| Single | 121 (24.1) | 61 (28.0) | 60 (21.1) |
| Divorced | 65 (12.9) | 31 (14.2) | 34 (12.0) |
| Widowed | 32 (6.4) | 8 (3.7) | 24 (8.5) |
| **Education[b]** | | | |
| <10 years | 205 (40.8) | 54 (24.8) | 151 (53.4) |
| ≥10 years | 292 (58.2) | 160 (73.4) | 132 (46.5) |
| **Occupation** | | | |
| Employed | 315 (62.7) | 160 (73.4) | 155 (54.6) |
| Retired | 124 (24.7) | 32 (14.7) | 92 (32.4) |
| Unemployed | 31 (6.2) | 7 (3.2) | 24 (8.5) |
| Other | 32 (6.4) | 19 (8.7) | 13 (4.6) |
| **Region of Portugal** | | | |
| North | 167 (33.3) | 74 (33.9) | 93 (32.7) |
| Center | 111 (22.1) | 45 (20.6) | 66 (23.2) |
| South | 224 (44.6) | 99 (45.4) | 125 (44.0) |
| **BMI, kg/m² Mean(SD)** | 26.9 (4.7) | 26.4 (5.2) | 27 (4.3) |
| **BMI classification** | | | |
| Healthy weight (18.5 – 24.9 kg/m²) | 197 (39.2) | 101 (46.3) | 96 (33.8) |
| Overweight (25–29.9 kg/m²) | 188 (37.5) | 70 (32.1) | 118 (41.5) |
| Obesity (≥30 kg/m²) | 117 (23.3) | 47 (21.6) | 70 (24.6) |
| **Smoking** | | | |
| Never/Ex smoker | 406 (80.9) | 179 (82.1) | 227 (79.9) |
| Current smoker | 96 (19.1) | 39 (17.9) | 57 (20.1) |
| **Smartphone ownership** | 383 (76.3) | 198 (90.8) | 185 (65.1) |
| **Health and Fitness apps use[c]** | 245 (48.8) | 129 (59.2) | 116 (40.8) |
| **Number of Comorbidities, Median [P25-P75]** | 2.0 [1.0-3.0] | 2.0 [1.0-3.0] | 2.0 [1.0-4.0] |
| **Exacerbations last year** | 205 (40.8) | 98 (45.0) | 107 (37.7) |
| **Unscheduled appointments with specialist in the last year** | 45 (9.0) | 26 (11.9) | 19 (6.7) |
| **CARAT classification** | | | |
| Controlled | 162 (32.3) | 65 (29.8) | 97 (34.2) |
| Uncontrolled | 340 (67.7) | 153 (70.2) | 187 (65.8) |
| **Mini-AQLQ questionnaire, Median (P25-P75)** | 5.2 (4.2-5.9) | 5.00 (4.1-5.8) | 5.3 (4.4-6.1) |

Data presented as n(%) unless otherwise indicated. AQLQ = Asthma Quality of Life Questionnaire; BMI = body mass index; CARAT = Control of Allergic Rhinitis and Asthma Test score; P25 = 25th percentile; P75 = 75th percentile; SD = Standard Deviation.

[a] 9 missing values,

[b] 5 missing values;

[c] 1 missing value

**Table 2. Crude and adjusted odds ratios (OR) explaining interest in Patient and Public Involvement Engagement and Participation (PPIEP).**

| | N | Crude OR | Adjusted OR |
|---|---|---|---|
| **Age groups** | | | |
| 18–64 years | 374 | 3.25 [2.03-5.20] | 1.37 [0.78-2.41] |
| ≥65 years | 119 | Reference | Reference |
| **Sex** | | | |
| Male | 180 | Reference | Reference |
| Female | 322 | 1.54 [1.06-2.25] | 1.53 [1.00-2.33] |
| **Marital status** | | | |
| Married | 284 | Reference | |
| Single/ Divorced/ Widowed | 218 | 1.19 [0.84-1.70] | |
| **Education** | | | |
| <10 years | 205 | Reference | Reference |
| ≥10 years | 292 | 3.39 [2.30-4.99] | 2.29 [1.47-3.57] |
| **Occupation** | | | |
| Employed | 315 | Reference | |
| Retired/Unemployed/Other | 187 | 0.44 [0.30-0.64] | |
| **Region of Portugal** | | | |
| North | 167 | Reference | |
| Center | 111 | 0.86 [0.53-1.39] | |
| South | 224 | 0.99 [0.66-1.49] | |
| **BMI kg/m²** | | 0.96 [0.93-1.00] | |
| **BMI classification** | | | |
| Healthy weight (18.5 – 24.9 kg/m²) | 197 | Reference | |
| Overweight (25–29.9 kg/m²) | 188 | 0.56 [0.37-0.85] | |
| Obesity (≥30 kg/m²) | 117 | 0.64 [0.40-1.01] | |
| **Smoking updated** | | | |
| Never/Ex smoker | 406 | Reference | |
| Current smoker | 96 | 0.87 [0.55-1.36] | |
| **Smartphone ownership** | | | |
| No | 119 | Reference | Reference |
| Yes | 383 | 5.71 [3.32-9.81] | 4.63 [2.51- 8.55] |
| **Health and Fitness apps use** | | | |
| No | 257 | Reference | |
| Yes | 245 | 2.12 [1.48- 3.04] | |
| **Number of comorbidities** | 502 | 0.88 [0.79-0.97] | |
| **Exacerbations last year** | | | |
| 0 | 297 | Reference | |
| ≥1 | 205 | 1.38 [0.96-1.97] | |
| **Unscheduled appointments with specialist in the last year** | | | |
| 0 | 457 | Reference | |
| ≥1 | 45 | 1.89 [1.02-3.51] | |
| **CARAT classification** | | | |
| Controlled | 162 | Reference | |
| Uncontrolled | 340 | 1.22 [0.83-1.78] | |
| **Mini-AQLQ questionnaire** | 502 | 0.84 [0.72- 0.98] | 0.77 [0.64- 0.92] |

BMI=body mass index; CARAT = Control of Allergic Rhinitis and Asthma Test score; AQLQ = Asthma Quality of Life Questionnaire

among the Portuguese general population would be valuable for contextualizing these findings. For instance, a study conducted in Cardiff, UK, revealed that 71% of the general population expressed a desire to participate in healthcare research and teaching [25]. However, making direct comparisons is challenging, as PPIEP initiatives are more established in the UK. Nonetheless, these insights indicate significant potential for enhancing patient engagement in research initiatives within Portugal.

Having higher education was significantly associated with higher odds of being interested in PPIEP, consistent with findings from a UK study that analysed the demographic characteristics of participants involved in PPIEP [19]. The association between higher educational levels and interest in PPIEP could be attributed to increased health literacy and a better understanding of the potential impact that participation can have on personal health outcomes and research dynamics. However, there are several non-educational, degree-related variables, such as internal self-motivation and the medical-patient relationship, [26] that also could impact patient interest and were not addressed in our study. The researcher-participant relationship is also an important factor to consider in PPIEP [27]. Positive engagement dynamics and trust-building measures are essential for sustaining participant motivation and continued involvement throughout the research process [24]. There is also an established association between health literacy and educational level [28]. Furthermore, health literacy alone may be a stronger predictor of health outcomes than educational level alone [29], making health literacy a factor to consider while implementing PPIEP in patient care or health research. By making health information more accessible through public health services and government partnerships [30], providing physical resources like visual aids [31], and offering online courses (e.g., EPAP —European Patient Ambassador Programme), we can increase health literacy and, consequently, PPIEP adherence across all patients, regardless of their educational level. Future studies should evaluate educational level and non-educational factors that may impact patient participation in PPIEP.

Smartphone ownership among patients with asthma was also associated with greater interest in participating in PPIEP. Individuals with higher digital health literacy scores tend to have better self-management and participation in their own medical decisions, mental and psychological state and quality of life [32]. But we need to consider existing disparities in the adoption and utilization of these technologies, often leaving disadvantaged groups, particularly individuals with low education levels; low incomes, immigrants, further behind [33]. Effective interventions addressing poor digital health literacy included education/training and social support [32]. A study demonstrated that digital platforms could facilitate communication in PPIEP activities between patients and researchers, especially when participants are located in multiple countries [34]. According to past literature, access to smart devices is an expected association with higher compliance with PPIEP and patient engagement activities and could be particularly impactful for patients with severe asthma [35]. Future PPIEP studies should include patients in secondary care to better understand the real impact of smart devices on compliance among patients with severe asthma.

Although the association was borderline significant, being a female demonstrated an association with higher odds of being interested in PPIEP. In the UK, a survey among people involved in PPIEP revealed a gender disparity in engagement, with women being more represented than men [18]. This follows the trend that was generally observed in research participation, with females being more likely to participate in health research than males [19,36].

Regarding patients' quality of life, our findings revealed that a better quality of life with improved asthma symptom control, as assessed with mini-AQLQ questionnaire, was significantly associated with lower odds of being interested in PPIEP. This observation aligns with the expectation that individuals may seek involvement in PPIEP primarily to improve their quality of life [24]. Better perceived disease control may negatively impact patient interest in participating in PPIEP activities, possibly because they perceive that they would not benefit from such interventions [24]. We acknowledge that there are limited studies on symptom control and compliance levels in PPIEP research settings. More research is needed to evaluate whether patients with better disease control are indeed less likely to participate in co-research and PPIEP activities. This information will help us determine whether we should implement strategies to promote their participation in asthma-related PPIEP. It also suggests a potential gap in engagement strategies that warrants further exploration to understand motivations beyond immediate health concerns.

While our study provides valuable insights into the factors associated with patient interest in PPIEP, it also underscores the need to address potential barriers to participation, such as limited accessibility of engagement formats (e.g., rigid meeting times); lack of compensation for participation; unfamiliarity with research processes.[24,37] Ensuring diverse and representative involvement in PPIEP is crucial for enhancing the translation of research into practice and promoting equitable healthcare. Future research should focus on identifying barriers faced by underrepresented groups and on the effectiveness of targeted inclusion strategies to foster a more inclusive approach to PPIEP. Approaches warranting investigation include community-partnered recruitment through trusted local organizations, development of plain-language materials, and implementation of flexible participation modalities (both virtual and in-person).[38]

We should acknowledge some strengths and limitations of this study. A key strength is its multicentre design, which involved data collection from various primary healthcare centres across all regions of Portugal, leading to a more diverse and representative sample. However, our primary care-based recruitment strategy may have limited our ability to capture perspectives from patients with severe asthma, as these individuals are more likely to be managed in secondary care settings. As this was a secondary analysis, our study was limited to the available sample from the main study, which included only adult participants. We recognize this as an important limitation and agree that future research investigating PPIEP interest and implementation strategies should actively include paediatric populations and their caregivers to ensure more comprehensive representation. In addition, a proportion of patients with asthma possibly declined participation at the invitation stage (stage 0), which may also limit the representativeness of our findings. We gathered a comprehensive set of individual-level characteristics, enabling us to explore the influence of various sociodemographic factors, smartphone accessibility, and cofactors such as quality of life and asthma control. Nonetheless, we did not take asthma severity or disease duration into account, which could potentially impact the predisposition to participate in PPIEP initiatives. Another limitation is that multicollinearity between predictors was not assessed, which could affect the interpretation of regression results.

## Conclusion

Two out of five patients with asthma were interested in participating in PPIEP initiatives. Factors such as higher levels of education, smartphone ownership, and female gender were associated with increased interest in PPIEP, while experiencing better quality of life was associated with reduced interest. By understanding participant characteristics, we can develop strategies to reach a wider audience and make PPIEP more inclusive and appealing. This, in turn, will lead to more diverse perspectives in research. Future studies should include a more comprehensive analysis of participant characteristics and their relationship with PPIEP involvement.

## Acknowledgments

We thank all the patients, carers, and citizens which participated in Epi-Asthma study. EPI-ASTHMA Study Group: Alice Nembrode Pereira Martins, Ana Barros, Ana Carrapato, Ana Catarina Pereira, Ana Catarina Ribeiro, Ana Coroas, Ana Cunha, Ana Filipa Ventura Pereira, Ana Gabriela Fernandes Peixoto Martins, Ana Isabel Barbosa, Ana Logrado, Ana Luísa Rodrigues, Ana Luís Marques Pinto Faria, Ana Mafalda Martins Oliveira Cunha, Ana Magalhães, Ana Margarida Correia, Ana Paula Romualdo, Ana Raquel Ribeiro, Ana Rita Correia, Ana Rita Faustino, Ana Rita Laranjeiro, Ana Rute Carreira, Ana Sofia Pinheiro, Ana Sofia Simões Carneiro de Oliveira, Ana Soares Jorge, Ana Teresa Frois, André Filipe Pereira Almeida Nazaré, Andreia Filipe, Andreia Mendes, António Rui Carvalho Moreira Lobo, Armando Brito de Sá, Bárbara Ferreira, Beatriz Henriques Antunes, Beatriz Palmira Freitas Alcântara, Benedita Graça Moura, Bernardo Pernardas, Bernardo Tomás Ferreira, Bruna Catarina Paiva Martins, Bruno Rei, Carla Joana Sousa, Carla Lopes, Carlos Nogueira, Carina de Almeida, Carolina Fernandes, Carolina Roldão, Catarina Baía, Catarina Lopes Pinheiro, Catarina Novais, Catarina Pinhão, Catarina Rosa, Catarina Sofia Cardoso Maduro, Cátia Chão, Cátia Patrícia Silva Cruz, Cecilia Teixeira, Cília

Nogueira, Clara Ferreira, Cláudia Alexandra Alves, Cláudia Lourenço, Daniel Casas, Daniel Castro, Daniela Morais, David Neves, Delfim Paulo Coelho Teixeira, Denise Varela, Diana Dias da Silva, Diana Palma, Diana Pacheco, Diana Tomaz, Diogo Maduro Pereira, Diogo Pereira, Diogo Rodrigues, Diogo Vieira Phalempin Cardoso, Dyna Torrado, Emanuel Noivo, Eurico Silva, Filipa Baptista, Filipa Figueiredo, Filipa Gonçalves, Filipa Lourenço, Flávia Ferreira, Francisca Isabel Ribeiro Caetano, Gabriel Teixeira, Gabriela Guiomar, Gabriela Montenegro, Gil André Espírito Santo Silva Santos, Graciete Denise do Carmo Barreto Santos, Guilherme de Amaral Mendes, Gustavo Santos, Helena Fonseca, Inês Diaz, Inês Filipa Garcia Moreira, Inês Freitas, Inês Isabel Sampaio de Macedo, Inês Matos, Inês Mendes, Inês Pinto, Inês Torrinha Martins Leão, Inês Varejão, Isabel Chão, Isabel Fragoso, Isabel Loureiro, Isabel Nuno, Joana Atabão, Joana Cameira, Joana Gonçalves, Joana Pinheiro, Joana Pimenta Falcão Marques, João António Castro e Silva, João Figueiredo, João Machado, João Marques, João Miguel Souto Henrique, João Nascimento, João Pedro Caixinhas Moreira Soares, João Pedro de Castro da Silva Alves, João Pedro Marques Ribeiro, João Ribeirinho Marques, João Ribeiro, João Sousa, Joaquim Santos, José Cabanas Carvalho, José Miguel Alvarez, Júlia Neves, Laura Fortuna, Leonor Carrapatoso, Lisandra Daniela Esteves Goncalves, Luís Pinheiro, Luís António Queirós, Luís Oliveira Soares, Luísa Rodrigues, Lurdes Matos, Madalena Barata Santos, Mafalda Jordão Cunha Abreu, Márcio Pereira, Margarida Espanhol, Margarida Magalhães, Margarida Vilarinho, Maria Francisca Silva, Maria Teresa Figueiral da Silva Pereira Leite, Mariana Bastos, Mariana Castro, Mariana José Figueira Almeida e Silva, Mariana Marques, Mariana Mateus Nunes Prudente, Marina Baptista, Marina Rodrigues, Marisa Abreu, Marisa Alves, Mário Machado Cruz, Miguel Cabanelas, Miguel Machado, Mónica Albuquerque, Mónica Reis, Nélson Carvalho, Nuno Jacinto, Nuno Ricardo Pina Soares, Olga Carneiro, Paulo André da Costa Lucas, Paulo Baptista Coelho, Patrícia Coelho de Azevedo, Pedro Aires, Pedro Costa Dias, Pedro Henrique Castro de Azevedo, Pedro Miguel de Moura Junqueira, Pedro Parreira da Silva, Pilar Marquez, Priscila Araújo, Rafael Dinis, Rafael Lopes da Cunha, Raquel Fatima Ferrao Andrade, Raquel Pessanha Santos, Rita Coutinho, Rita Paraíso, Rita Sousa, Rizério Salgado, Rui Garcia, Salomé Vieira Costa e Silva, Salomé Carvalho, Sandrine Isabelle Dias, Sara Alexandra Ramalho Pinheiro, Sara Carmona, Sara Fernandez, Sara Isabel Calmeiro Pinto, Sara Raquel Catarino Almeida Ribeiro Braga, Sara Rodrigues da Silva, Sara Santana, Sara Santos, Serenela Morgado Ventura da Luz, Sofia Andreia Pimenta Diogo, Sofia Figueiras, Sofia Lima, Sofia Jardim, Sónia Silva, Soraia Raquel Machado Vaz Osorio, Susana Corte-Real, Teresa Costa Campos, Teresa Libório, Teresa Pascoal, Teresa Raposo, Tiago André Oliveira Guimarães de Matos, Tiago Antunes, Vanessa Costa, Vanessa Horta, Vera Sousa, Vítor Rego, Viviana Barreira.

## Author contributions

**Writing – original draft:** Mafalda Simões Cunha, João da Silva Santos.

**Writing – review & editing:** Mafalda Simões Cunha, João da Silva Santos, Ana Sá-Sousa, Ana Luísa Neves, Ana Alves da Silva, Diana Gomes, Janete Quelhas Santos, Filipa Bernardo, Jaime Correia de Sousa, João Almeida-Fonseca, Cristina Jácome.

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
