## [Decision Letter · Decision Letter 0]

14 Mar 2025

Dear Dr. Jácome,

Thank you for submitting your manuscript to PLOS ONE. After careful consideration, we feel that it has merit but does not fully meet PLOS ONE’s publication criteria as it currently stands. Therefore, we invite you to submit a revised version of the manuscript that addresses the points raised during the review process.

Dear Authors,

We hope this message finds you well.

During the evaluation of your work, we identified the need for some corrections based on the reviewers' observations. In order to proceed with the evaluation process, we kindly request that the necessary modifications be made.

Please ensure that all changes are **highlighted in yellow**  in the document to facilitate the identification of the revisions made.

We look forward to receiving the revised version to continue the evaluation.

Best regards,

We look forward to receiving your revised manuscript.

Kind regards,

Manuela Mendonça Figueirêdo Coelho, Ph.D

Academic Editor

PLOS ONE

Journal Requirements:

“Filipa Bernardo (FB) is an employee of AstraZeneca, Produtos Farmacêuticos SA. The remaining authors have no conflicts of interest to declare.”

We note that one or more of the authors are employed by a commercial company: AstraZeneca

Reviewers' comments:

Reviewer's Responses to Questions

**Comments to the Author**

1. Is the manuscript technically sound, and do the data support the conclusions?

Reviewer #1: Yes

Reviewer #2: Partly

Reviewer #3: Yes

2. Has the statistical analysis been performed appropriately and rigorously?

Reviewer #1: Yes

Reviewer #2: Yes

Reviewer #3: Yes

3. Have the authors made all data underlying the findings in their manuscript fully available?

Reviewer #1: No

Reviewer #2: No

Reviewer #3: Yes

4. Is the manuscript presented in an intelligible fashion and written in standard English?

Reviewer #1: Yes

Reviewer #2: Yes

Reviewer #3: Yes

Reviewer #1: Cunha and colleagues sought to identify the factors associated with the interest in being involved in a Patient and Public Involvement Engagement and Participation (PPIEP) among patients with asthma. It is a secondary analysis of the EPI-ASTHMA nationwide asthma prevalence study where a total of 502 participants with asthma (52 ± 15.9 years, 64.1% females) from 38 primary healthcare centers from mainland Portugal were analyzed. They found that a total of 218 (43.4%) participants with asthma were interested in being involved in the Patient and Public Involvement Engagement and Participation (PPIEP). They concluded that two out of five patients with asthma were interested in participating in PPIEP initiatives. They also indicated that by understanding patient characteristics willing to participate in PPIEP, they could develop strategies to reach a wider audience and make PPIEP more inclusive. The identified associated factors that they observed need to be further investigated.

This is an interesting report based on an effort to analyze willingness of patients to participate in asthma research. I have a few suggestions to improve the presentation along with a few minor comments for corrections to the text:

1. Title – if necessary, you could eliminate the Patient and Public Involvement Engagement and Participation portion.

2. Abstract, p. 2, line 54 – do you have specific strategies in mind that could be included?

3. Introduction – p. 3 – line 72 - was the Patient and Public Involvement Engagement and Participation initiative set up for one study or for many studies? Perhaps a little more background on the development of this program would be helpful. How long has it been going on? Has it been successful? Will it continue?

4. Introduction, p. 4, lines 104 to 105 – what is the goal of this program after having collected this information? In what manner will it continue?

5. Study Design, p.4 – a little more information about the Patient and Public Involvement Engagement and Participation program would be helpful in terms of the scope of the program, target age, entry criteria, exclusion criteria and also the anticipated burden of the study PPIEP is recruiting for and whether that was included in the invitation to particpate. What were potential participants told would be done with their data and the reason for providing their permission?

6. Participants, p.4 – What was the target population surveyed? Were pediatric patients purposely excluded? Was there data collected on their prior participation in asthma or any other studies?

7. Discussion, p. 10, line 216 – what is the description of the disadvantaged population in Portugal? Is it low-income families, homeless, a particular race/ethnicity?

8. Discussion, p. 10, line 242 – What would you consider an important level of diversity and representative population for an asthma study in Portugal? Same is true for p. 11, line 244, what is the underrepresented groups in Portugal? That could be different from other countries, such as the United States.

9. Discussion, p. 11, line 249 – What is secondary care patients?

10. Discussion, p. 11 – it might be of interest to compare your population findings to published studies on the various levels of asthma severity.

11. Discussion, p.11, line 260 – What is your future vision for the application of this data and meeting your goals for studies? Will PPiEP continue its work and for what purpose?

12. Discussion, Conclusions, p. 11, line 266 – What strategies do you propose?

13. Discussion, Conclusions, p. 11, line 267 – How do you define a diverse perspective for your purposes in conducting asthma studies?

14. References – Check to see that your author format for references is consistent with journal requirements.

Minor corrections:

1. Page 9, line 201- ….did not address…

2. Page 10, line 215 - ….to be cautious regarding…

Reviewer #2: Your study is interesting and beneficial to those interested in patient recruitment. I think it should be published. Your enrollment and demographics are quite similar to what our group found when we investigated participation in survey-based research during the COVID pandemic in the US.

Because the EPI-ASTHMA study is still in press, I am unable to find all of the details related to recruitment (which is why I said details were not available).

For the most part, the research and conclusions are sound. However, one obvious source of bias is that you recruited participants by phone. I think the reason smartphone ownership was so strongly tied to participation was because of the recruitment method. The recruitment method could also introduce bias for other factors such as age, gender, degree of asthma control, etc. However, smartphone usage is the most obvious. This bias definitely needs to be mentioned in your discussion. And in fact, one could argue that smartphone ownership should not be included in your analysis at all.

I noticed that the number of participants in the study was much less than expected. Without access to the EPI-ASTHMA manuscript I do not know why, but this may be worth mentioning. There was also no explanation as to why 16 participants were excluded from your analysis.

There were grammar/syntax errors in lines 134, 201, and 202. THe sentence starting on line 215 does not make sense and needs to be reworked.

Reviewer #3: Dear Authors,

Thank you for your efforts. There are some major points that need improvement:

Title:

The title would benefit from reducing the words. Some words like “factors” are vague and better not to be used in title.

Abstract:

Line 45: “total of 502 participants with asthma (52 ± 15.9 years, 64.1% females)”: Usually these data are described in result not method.

Line 47: A multivariable logistic regression model was built: Please identify what variable the model was built for.

Introduction:

The research aim is clear but brief. Consider adding 1-2 sentences about the significance of identifying these factors and potential implications.

Method:

- The method section is a bit unclear. It is not clearly mentioned if the study is a secondary analysis of data available, or they have actively gathered data. For example, they have stated that they have gotten consent from the participants, meaning active participation. However, in all details they are talking about a third party data gathering refereeing other articles.

- The key variable about PPIEP interest is not well described. Add details about how this was measured.

- Consider adding information about how you handled missing data

- Specify which variables were tested in the simple logistic regressions

- Consider mentioning how you assessed for multicollinearity in your regression models

Result:

- Line 166-171: When reporting the multivariable results, consider adding p-values alongside the odds ratios and confidence intervals.

- The interpretation of OR 1.53 (95%CI 1.00-2.33) for females should be approached with caution as the lower bound of the confidence interval is exactly 1.00, suggesting borderline significance.

- There is a problem in Smartphone ownership data percentage. 198 out of 218 would be 90.8%, not 93.6%. Please check all data again.

Discussion

Try organizing paragraphs more thematically.

Some minor editions:

1. In line 63, “This variation” could be clearer by specifying “This variation in asthma control”.

2. I am not sure whether PCC stands for primary healthcare centres. (Line 110)

3. Line 110: “Data was accessed” should be “Data were accessed”.

4. More clearly define “current asthma” versus “possible asthma” or reference where these definitions can be found.

5. Line 192: “more women than man” shou

**Do you want your identity to be public for this peer review?** For information about this choice, including consent withdrawal, please see our Privacy Policy

Reviewer #1: **Yes: ** Stanley J. Szefler, MD

Reviewer #2: No

Reviewer #3: **Yes: ** Noosha Samieefar

---

## [Author Response · Author response to Decision Letter 1]

3 Apr 2025

Dear Prof. Manuela Mendonça Figueirêdo Coelho,

Academic Editor of PLOS One,

March 26th, 2025

Title: Interest in public involvement, engagement and participation in research among patients with asthma: A secondary analysis of the EPI-ASTHMA study

Manuscript ID: PONE-D-25-04695

Dear Prof. Manuela Mendonça Figueirêdo Coelho,

Thank you for considering our manuscript. Below you will find a point-by-point reply to the issues raised during the review process. We would like to thank you, and the reviewers for the useful comments, which helped us to improve the manuscript significantly. All changes made in the manuscript are highlighted.

I am looking forward to hearing from you.

Yours sincerely,

Cristina Jácome

Academic Editor

Response: The manuscript has been modified to meet PLOS ONE style, namely, abstract headings were removed, manuscript heading styles edited; title page edited.

“Filipa Bernardo (FB) is an employee of AstraZeneca, Produtos Farmacêuticos SA. The remaining authors have no conflicts of interest to declare.”

We note that one or more of the authors are employed by a commercial company: AstraZeneca

Please also include the following statement within your amended Funding Statement. “The funder provided support in the form of salaries for authors [insert relevant initials], but did not have any additional role in the study design, data collection and analysis, decision to publish, or preparation of the manuscript. The specific roles of these authors are articulated in the ‘author contributions’ section.” If your commercial affiliation did play a role in your study, please state and explain this role within your updated Funding Statement.

Response: The Funding Statement and Competing Interests Statement were updated and added in the Cover Letter as requested. Please see below:

Funding: This study was sponsored and funded by AstraZeneca. The funder provided support in the form of salaries for authors FB, but did not have any additional role in the study design, data collection and analysis, decision to publish, or preparation of the manuscript.

Conflict of interest: We have read the journal’s policy and the authors of this manuscript have the following competing interests: FB is an employee of AstraZeneca, Produtos Farmacêuticos SA. FB does not have any other conflicts of interest to declare. This does not alter our adherence to PLOS ONE policies on sharing data and materials. The remaining authors have no conflicts of interest to declare.

3. We note that you have indicated that there are restrictions to data sharing for this study. For studies involving human research participant data or other sensitive data, we encourage authors to share de-identified or anonymized data. However, when data cannot be publicly shared for ethical reasons, we allow authors to make their data sets available upon request. For information on unacceptable data access restrictions, please see http://journals.plos.org/plosone/s/data-availability#loc-unacceptable-data-access-restrictions

Response: The Data Availability statement has been updated: “There are ethical or legal restrictions on sharing a de-identified data set, because that the data contain potentially identifying or sensitive patient information and the ethics committee imposed them. This study was approved by the ethics committees of the Regional Health Administrations of North (CE/2022/117), Center (27/2021), Lisbon and Tagus Valley (2775/CES/2022), Alentejo (11/CE/2022) and Algarve (1/2022) and of the Local Health Units of Matosinhos (38/CES/JAS) and Alto Minho (38/2021). The original approval from the Ethical Review Boards did not include direct and free data access and therefore, data cannot be made freely available. All data supporting the findings are available from the Faculty of Medicine of University of Porto for all interested researchers who meet the criteria for access to confidential data. Please contact Cristina Jácome, the corresponding author at cjacome@med.up.pt, or the Faculty of Medicine of University of Porto at fmup@med.up.pt to request the data.

I would also like to stress that a similar Data Availability Statement was published in a previous secondary analysis also published in PLOS One (https://journals.plos.org/plosone/article?id=10.1371/journal.pone.0317614).

Reviewer #1

Cunha and colleagues sought to identify the factors associated with the interest in being involved in a Patient and Public Involvement Engagement and Participation (PPIEP) among patients with asthma. It is a secondary analysis of the EPI-ASTHMA nationwide asthma prevalence study where a total of 502 participants with asthma (52 ±15.9 years, 64.1% females) from 38 primary healthcare centers from mainland Portugal were analyzed. They found that a total of 218 (43.4%) participants with asthma were interested in being involved in the Patient and Public Involvement Engagement and Participation (PPIEP). They concluded that two out of five patients with asthma were interested in participating in PPIEP initiatives. They also indicated that by understanding patient characteristics willing to participate in PPIEP, they could develop strategies to reach a wider audience and make PPIEP more inclusive. The identified associated factors that they observed need to be further investigated. This is an interesting report based on an effort to analyze willingness of patients to participate in asthma research. I have a few suggestions to improve the presentation along with a few minor comments for corrections to the text:

1. Title – if necessary, you could eliminate the Patient and Public Involvement Engagement and Participation portion.

Response: The title has been reduced, but the expression “Patient and Public Involvement, Engagement, and Participation,” as defined by the UK National Institute for Health and Care Research, has been retained. This term was recently defined to replace previous PPI and PPIE and unfortunately it is still relatively unfamiliar to the general scientific community, and ongoing efforts are required to enhance its recognition and understanding. Please see lines 1-2: “Interest in public involvement, engagement and participation in research among patients with asthma: A secondary analysis of the EPI-ASTHMA study”

2. Abstract, p. 2, line 54 – do you have specific strategies in mind that could be included?

Response: While we have included examples of specific strategies in the Discussion section (lines 237-246), we believe it would not be appropriate to add them to the abstract's conclusion. These suggestions are preliminary and were not explicitly discussed or validated by the study participants, so presenting them in the abstract could overstate their significance. We are happy to clarify further or discuss alternative adjustments if needed.

3. Introduction – p. 3 – line 72 - was the Patient and Public Involvement Engagement and Participation initiative set up for one study or for many studies? Perhaps a little more background on the development of this program would be helpful. How long has it been going on? Has it been successful? Will it continue?

Response: In lines 72-73 we are defining PPIE in a broad way, regardless of the chronic condition using the most recent definition of the UK National Institute for Health and Care Research (NIHR).In the subsequent lines (74-80), we are trying to show to the reader that recent evidence from original and synthesis studies(ref 10, 11 and 12) demonstrate that PPIE in health research has important advantages. From lines 80-83, we point out the current challenges in implementing PPIE.

All this evidence came from different studies and not from a single experience or program. We tried to make that clear. Please see lines 66-73: “A review of reviews showed that incorporating PPIEP into health research and healthcare can be a powerful tool for identifying patient needs and optimising treatment approaches [11]. PPIEP aims to improve patient outcomes and reduce healthcare costs by empowering patients to make informed decisions and promoting open discussions about disease symptoms and comorbidities [12]. Yet, previous experiences in other context or conditions demonstrate that the effectiveness of PPIEP depends on various factors, including the quality of the relationship between researchers, patients, and the public [13], accessibility of information and training opportunities [14], as well as the health limitations and sociodemographic characteristics of the participants [15].”

4. Introduction, p. 4, lines 104 to 105 – what is the goal of this program after having collected this information? In what manner will it continue?

Response: The current paper does not address the next steps. The current paper focuses on raising awareness that patients are interested in PPIEP initiatives related to asthma. But it also points out that we are captivating only patients with certain characteristics. If broader patient involvement is desired in the future, alternative strategies will be needed to reach and engage a more diverse group.

5. Study Design, p.4 – a little more information about the Patient and Public Involvement Engagement and Participation program would be helpful in terms of the scope of the program, target age, entry criteria, exclusion criteria and also the anticipated burden of the study PPIEP is recruiting for and whether that was included in the invitation to particpate. What were potential participants told would be done with their data and the reason for providing their permission?

Response: Thank you for your valuable comment. We would like to clarify that at the time of this study, no formal PPIEP program existed. The research team conducted this work specifically to assess the potential feasibility of establishing such a network in future. We have now included additional details about the specific question posed to participants to assess their interest. Please see page lines 129-133: “At the end of the stage 2 visit, patients answered a question regarding PPIEP interest (‘We intend to constitute a group of people with asthma with a more active role in research. Your participation will be occasional and will contribute to enrich the research projects - from their conception to the final result - with the perspective and needs of people who deal with asthma day by day. Are you interested in collaborating with this group?’).”.

6. Participants, p.4 – What was the target population surveyed? Were pediatric patients purposely excluded? Was there data collected on their prior participation in asthma or any other studies?

Response: Thank you for your comment. We would like to clarify that the assessment of PPIEP interest was conducted among all adult patients with asthma during stage 2 of the Epi-asthma study. As this is a secondary analysis of Epi-asthma—a study primarily designed to assess adult asthma prevalence in Portugal—our findings are inherently limited to the adult population. Pediatric patients were not purposely excluded, yet we acknowledge this as a limitation of our research, which we have now explicitly addressed in our discussion section. Please see lines 251-255: “As this was a secondary analysis, our study was limited to the available sample from the main study, which included only adult participants. We recognize this as an important limitation and agree that future research investigating PPIEP interest and implementation strategies should actively include paediatric populations and their caregivers to ensure more comprehensive representation.”

7. Discussion, p. 10, line 216 – what is the description of the disadvantaged population in Portugal? Is it low-income families, homeless, a particular race/ethnicity?

Response: We have now clarified this. Please see lines 210-212: “But we need to consider existing disparities in the adoption and utilization of these technologies, often leaving disadvantaged groups, particularly individuals with low education levels; low incomes, immigrants, further behind [34].”

8. Discussion, p. 10, line 242 – What would you consider an important level of diversity and representative population for an asthma study in Portugal? Same is true for p. 11, line 244, what is the underrepresented groups in Portugal? That could be different from other countries, such as the United States.

Response: We have clarified above what are the common underrepresented groups in Portugal. In this paragraph we have tried to provide more information on the known barriers to participation by these groups and also highlighted the need to gather evidence of specific inclusion strategies. Please see lines 237-246: “While our study provides valuable insights into the factors associated with patient interest in PPIEP, it also underscores the need to addre

---

## [Editor Report · Decision Letter 1]

27 Apr 2025

Dear Dr. Jácome,

Thank you for submitting your manuscript to PLOS ONE. After careful consideration, we feel that it has merit but does not fully meet PLOS ONE’s publication criteria as it currently stands. Therefore, we invite you to submit a revised version of the manuscript that addresses the points raised during the review process.

**Dear Authors,**

I would like to sincerely congratulate you on the excellent work carried out in revising the manuscript “Interest in Public Involvement, Engagement and Participation in Research among Patients with Asthma: A Secondary Analysis of the EPI-ASTHMA Study.”

After a careful and detailed review, I confirm that all the reviewers' comments and editorial requirements have been fully addressed. The revisions have made the manuscript much clearer, more consistent, and well-aligned with PLOS ONE’s guidelines.

As an optional suggestion, aiming only to further strengthen the final submission, I would like to propose **a few minor improvements** :

**Multicollinearity assessment** : Although the absence of multicollinearity evaluation was acknowledged in the response letter, it would be advisable to briefly mention this in the manuscript’s Limitations section. A sentence such as the following could be added:“Another limitation is that multicollinearity between predictors was not assessed, which could affect the interpretation of regression results.”**Consistency of data collection dates** : There is a minor inconsistency regarding the final date of data collection (December 2023 vs. February 2024). I recommend verifying and ensuring consistency throughout the manuscript.**Title formatting** : Following PLOS ONE formatting style, please consider capitalizing the main words in the title as follows:Interest in Public Involvement, Engagement and Participation in Research among Patients with Asthma: A Secondary Analysis of the EPI-ASTHMA Study.

These are only small and non-mandatory recommendations, focused mainly on stylistic refinement and methodological transparency. I emphasize that the manuscript already demonstrates a high technical and scientific standard, and that your careful work throughout the revision process is clearly reflected in the final version.

Congratulations once again to the entire team for the commitment and dedication. I remain available for any further clarifications if needed.

Best regards,

We look forward to receiving your revised manuscript.

Kind regards,

Manuela Mendonça Figueirêdo Coelho, Ph.D

Academic Editor

PLOS ONE
---

## [Author Response · Author response to Decision Letter 2]

28 Apr 2025

Dear Prof. Manuela Mendonça Figueirêdo Coelho,

Academic Editor of PLOS One,

April 28th, 2025

Title: Interest in Public Involvement, Engagement and Participation in Research among Patients with Asthma: A Secondary Analysis of the EPI-ASTHMA Study

Manuscript ID: PONE-D-25-04695

Dear Prof. Manuela Mendonça Figueirêdo Coelho,

Thank you for considering our manuscript. Below you will find a point-by-point reply to the issues raised. We would like to thank you for the useful comments. All changes made in the manuscript are highlighted.

I am looking forward to hearing from you.

Yours sincerely,

Cristina Jácome

Academic Editor

As an optional suggestion, aiming only to further strengthen the final submission, I would like to propose

a few minor improvements:

1. Multicollinearity assessment: Although the absence of multicollinearity evaluation was acknowledged in the response letter, it would be advisable to briefly mention this in the manuscript’s Limitations section. A sentence such as the following could be added:

“Another limitation is that multicollinearity between predictors was not assessed, which could affect the interpretation of regression results.”

Response: Discussion of this limitation has been added. Please see lines 261-263.

2. Consistency of data collection dates: There is a minor inconsistency regarding the final date of data collection (December 2023 vs. February 2024). I recommend verifying and ensuring consistency throughout the manuscript.

Response: Data collection closed in March 2024. This has now been clarified. Please see line 101.

3.Title formatting: Following PLOS ONE formatting style, please consider capitalizing the main words in the title as follows:

Interest in Public Involvement, Engagement and Participation in Research among Patients with Asthma: A Secondary Analysis of the EPI-ASTHMA Study.

Response: Title has been edited as suggested.

---

## [Editor Report · Decision Letter 2]

12 May 2025

Dear Dr. Jácome,

Thank you for submitting your manuscript to PLOS ONE. After careful consideration, we feel that it has merit but does not fully meet PLOS ONE’s publication criteria as it currently stands. Therefore, we invite you to submit a revised version of the manuscript that addresses the points raised during the review process.

Dear authors,

If you have made the corrections requested by the reviewers below, please send a new file with the corrections marked in blue, as well as a response letter regarding each correction.

Thank you

We look forward to receiving your revised manuscript.

Kind regards,

Manuela Mendonça Figueirêdo Coelho, Ph.D

Academic Editor

PLOS ONE

Journal Requirements:

Additional Editor Comments:

Reviewer 01:

Cunha and colleagues sought to identify factors associated with interest in engaging in Patient Engagement and Participation and Public Involvement (PEIEP) among patients with asthma. This is a secondary analysis of the EPI-ASTHMA national asthma prevalence study, which included a total of 502 participants with asthma (52 ± 15.9 years, 64.1% female) from 38 primary health care centers in mainland Portugal. They found that a total of 218 (43.4%) participants with asthma were interested in engaging in PEIEP. They concluded that two out of five patients with asthma were interested in participating in PEIEP initiatives. They also indicated that by understanding the characteristics of patients willing to participate in PEIEP, they could develop strategies to reach a wider audience and make PEIEP more inclusive. The associated factors they identified need to be investigated further. This is an interesting report based on an effort to analyze patient willingness to participate in asthma research. I have a few suggestions for improving the presentation, along with some minor comments for corrections in the text: 1. Title - if necessary, you could eliminate the Patient Engagement and Participation and Public Involvement section. 2. Abstract, p. 2, line 54 - do you have specific strategies in mind that could be included? 3. Introduction - p. 3 - line 72 - was the Patient Engagement and Participation and Public Involvement initiative set up for one study or for many studies? Perhaps a little more background on the development of this program would be helpful. How long has it been going on? Has it been successful? Will it continue? 4. Introduction, p. 4, lines 104-105 - what is the goal of this program once you have collected this information? How will it continue? 5. Study Design, p.4 – a little more information about the Patient and Public Engagement and Participation programme would be helpful in terms of scope of the programme, target age, entry criteria, exclusion criteria and also the anticipated burden of the study for which PPIEP is recruiting and whether this was included in the invitation to participate. What were potential participants told would be done with their data and the reason for providing their permission? 6. Participants, p.4 – Who was the target population studied? Were paediatric patients purposefully excluded? Was there data collected on their previous participation in asthma or any other studies? 7. Discussion, p. 10, line 216 – what is the description of the disadvantaged population in Portugal? Is it low income families, homeless, a particular race/ethnicity? 8. Discussion, p. 10, line 242 – What would you consider an important level of diversity and representative population for an asthma study in Portugal? The same is true for p. 11, line 244, which groups are underrepresented in Portugal? This may be different from other countries, such as the United States. 9. Discussion, p. 11, line 249 - What are secondary care patients? 10. Discussion, p. 11 - It may be interesting to compare your population findings with published studies on the various levels of asthma severity. 11. Discussion, p. 11, line 260 - What is your future vision for applying these data and achieving your study objectives? Will PPiEP continue its work and for what purpose? 12. Discussion, Conclusions, p. 11, line 266 - What strategies do you propose? 13. Discussion, Conclusions, p. 11, line 267 - How do you define a diverse perspective for your purposes in conducting asthma studies? 14. References - Please check that your author's format for references is consistent with the journal's requirements. Minor corrections: 1. Page 9, line 201 - ....did not address... 2. Page 10, line 215 - ....to be cautious about...

Reviewer 02

Your study is interesting and beneficial to those interested in patient recruitment. I think it should be published. Your enrollment and demographics are quite similar to what our group found when we investigated participation in survey-based research during the COVID pandemic in the US. Since the EPI-ASTHMA study is still in the works, I am unable to find all the details related to recruitment (which is why I said the details were not available). For the most part, the research and conclusions are sound. However, one obvious source of bias is that you recruited participants by phone. I think the reason smartphone ownership was so strongly linked to participation was because of the recruitment method. The recruitment method may also introduce bias for other factors such as age, gender, degree of asthma control, etc. However, the use

Revisor 03

Dear Authors,

Thank you for your efforts. There are some major points that need improvement:

Title:

The title would benefit from reducing the words. Some words like “factors” are vague and better not to be used in title.

Abstract:

Line 45: “total of 502 participants with asthma (52 ± 15.9 years, 64.1% females)”: Usually these data are described in result not method.

Line 47: A multivariable logistic regression model was built: Please identify what variable the model was built for.

Introduction:

The research aim is clear but brief. Consider adding 1-2 sentences about the significance of identifying these factors and potential implications.

Method:

- The method section is a bit unclear. It is not clearly mentioned if the study is a secondary analysis of data available, or they have actively gathered data. For example, they have stated that they have gotten consent from the participants, meaning active participation. However, in all details they are talking about a third party data gathering refereeing other articles.

- The key variable about PPIEP interest is not well described. Add details about how this was measured.

- Consider adding information about how you handled missing data

- Specify which variables were tested in the simple logistic regressions

- Consider mentioning how you assessed for multicollinearity in your regression models

Result:

- Line 166-171: When reporting the multivariable results, consider adding p-values alongside the odds ratios and confidence intervals.

- The interpretation of OR 1.53 (95%CI 1.00-2.33) for females should be approached with caution as the lower bound of the confidence interval is exactly 1.00, suggesting borderline significance.

- There is a problem in Smartphone ownership data percentage. 198 out of 218 would be 90.8%, not 93.6%. Please check all data again.

Discussion

Try organizing paragraphs more thematically.

Some minor editions:

1. In line 63, “This variation” could be clearer by specifying “This variation in asthma control”.

2. I am not sure whether PCC stands for primary healthcare centres. (Line 110)

3. Line 110: “Data was accessed” should be “Data were accessed”.

4. More clearly define “current asthma” versus “possible asthma” or reference where these definitions can be found.

5. Line 192: “more women than man” shou

---

## [Author Response · Author response to Decision Letter 3]

23 May 2025

Dear Prof. Manuela Mendonça Figueirêdo Coelho,

Academic Editor of PLOS One,

May 23th, 2025

Title: Interest in Public Involvement, Engagement and Participation in Research among Patients with Asthma: A Secondary Analysis of the EPI-ASTHMA Study

Manuscript ID: PONE-D-25-04695

Dear Prof. Manuela Mendonça Figueirêdo Coelho,

Thank you for considering our manuscript. Below you will find a point-by-point reply to the issues raised. We would like to thank you for the useful comments. All changes made in the manuscript are highlighted in blue.

I am looking forward to hearing from you.

Yours sincerely,

Cristina Jácome

Third revision

Academic Editor

If you have made the corrections requested by the reviewers below, please send a new file with the corrections marked in blue, as well as a response letter regarding each correction.

Response: Something was probably lost in the process. In all interactions (R1 and R2) I have uploaded a point-by-point response letter, together with a manuscript version with changes highlighted. In this response letter, you can find all responses provided in these two interactions. Please see below. In addition, the manuscript is now provided with all corrections from these two interactions marked in blue.

Second revision

Academic Editor

As an optional suggestion, aiming only to further strengthen the final submission, I would like to propose

a few minor improvements:

1. Multicollinearity assessment: Although the absence of multicollinearity evaluation was acknowledged in the response letter, it would be advisable to briefly mention this in the manuscript’s Limitations section. A sentence such as the following could be added:

“Another limitation is that multicollinearity between predictors was not assessed, which could affect the interpretation of regression results.”

Response: Discussion of this limitation has been added. Please see lines 261-263.

2. Consistency of data collection dates: There is a minor inconsistency regarding the final date of data collection (December 2023 vs. February 2024). I recommend verifying and ensuring consistency throughout the manuscript.

Response: Data collection closed in March 2024. This has now been clarified. Please see line 101.

3.Title formatting: Following PLOS ONE formatting style, please consider capitalizing the main words in the title as follows:

Interest in Public Involvement, Engagement and Participation in Research among Patients with Asthma: A Secondary Analysis of the EPI-ASTHMA Study.

Response: Title has been edited as suggested.

First revision March 26th

Academic Editor

Response: The manuscript has been modified to meet PLOS ONE style, namely, abstract headings were removed, manuscript heading styles edited; title page edited.

“Filipa Bernardo (FB) is an employee of AstraZeneca, Produtos Farmacêuticos SA. The remaining authors have no conflicts of interest to declare.”

We note that one or more of the authors are employed by a commercial company: AstraZeneca

Please also include the following statement within your amended Funding Statement. “The funder provided support in the form of salaries for authors [insert relevant initials], but did not have any additional role in the study design, data collection and analysis, decision to publish, or preparation of the manuscript. The specific roles of these authors are articulated in the ‘author contributions’ section.” If your commercial affiliation did play a role in your study, please state and explain this role within your updated Funding Statement.

Response: The Funding Statement and Competing Interests Statement were updated and added in the Cover Letter as requested. Please see below:

Funding: This study was sponsored and funded by AstraZeneca. The funder provided support in the form of salaries for authors FB, but did not have any additional role in the study design, data collection and analysis, decision to publish, or preparation of the manuscript.

Conflict of interest: We have read the journal’s policy and the authors of this manuscript have the following competing interests: FB is an employee of AstraZeneca, Produtos Farmacêuticos SA. FB does not have any other conflicts of interest to declare. This does not alter our adherence to PLOS ONE policies on sharing data and materials. The remaining authors have no conflicts of interest to declare.

3. We note that you have indicated that there are restrictions to data sharing for this study. For studies involving human research participant data or other sensitive data, we encourage authors to share de-identified or anonymized data. However, when data cannot be publicly shared for ethical reasons, we allow authors to make their data sets available upon request. For information on unacceptable data access restrictions, please see http://journals.plos.org/plosone/s/data-availability#loc-unacceptable-data-access-restrictions

Response: The Data Availability statement has been updated: “There are ethical or legal restrictions on sharing a de-identified data set, because that the data contain potentially identifying or sensitive patient information and the ethics committee imposed them. This study was approved by the ethics committees of the Regional Health Administrations of North (CE/2022/117), Center (27/2021), Lisbon and Tagus Valley (2775/CES/2022), Alentejo (11/CE/2022) and Algarve (1/2022) and of the Local Health Units of Matosinhos (38/CES/JAS) and Alto Minho (38/2021). The original approval from the Ethical Review Boards did not include direct and free data access and therefore, data cannot be made freely available. All data supporting the findings are available from the Faculty of Medicine of University of Porto for all interested researchers who meet the criteria for access to confidential data. Please contact Cristina Jácome, the corresponding author at cjacome@med.up.pt, or the Faculty of Medicine of University of Porto at fmup@med.up.pt to request the data.

I would also like to stress that a similar Data Availability Statement was published in a previous secondary analysis also published in PLOS One (https://journals.plos.org/plosone/article?id=10.1371/journal.pone.0317614).

Reviewer #1

Cunha and colleagues sought to identify the factors associated with the interest in being involved in a Patient and Public Involvement Engagement and Participation (PPIEP) among patients with asthma. It is a secondary analysis of the EPI-ASTHMA nationwide asthma prevalence study where a total of 502 participants with asthma (52 ±15.9 years, 64.1% females) from 38 primary healthcare centers from mainland Portugal were analyzed. They found that a total of 218 (43.4%) participants with asthma were interested in being involved in the Patient and Public Involvement Engagement and Participation (PPIEP). They concluded that two out of five patients with asthma were interested in participating in PPIEP initiatives. They also indicated that by understanding patient characteristics willing to participate in PPIEP, they could develop strategies to reach a wider audience and make PPIEP more inclusive. The identified associated factors that they observed need to be further investigated. This is an interesting report based on an effort to analyze willingness of patients to participate in asthma research. I have a few suggestions to improve the presentation along with a few minor comments for corrections to the text:

1. Title – if necessary, you could eliminate the Patient and Public Involvement Engagement and Participation portion.

Response: The title has been reduced, but the expression “Patient and Public Involvement, Engagement, and Participation,” as defined by the UK National Institute for Health and Care Research, has been retained. This term was recently defined to replace previous PPI and PPIE and unfortunately it is still relatively unfamiliar to the general scientific community, and ongoing efforts are required to enhance its recognition and understanding. Please see lines 1-2: “Interest in public involvement, engagement and participation in research among patients with asthma: A secondary analysis of the EPI-ASTHMA study”

2. Abstract, p. 2, line 54 – do you have specific strategies in mind that could be included?

Response: While we have included examples of specific strategies in the Discussion section (lines 237-246), we believe it would not be appropriate to add them to the abstract's conclusion. These suggestions are preliminary and were not explicitly discussed or validated by the study participants, so presenting them in the abstract could overstate their significance. We are happy to clarify further or discuss alternative adjustments if needed.

3. Introduction – p. 3 – line 72 - was the Patient and Public Involvement Engagement and Participation initiative set up for one study or for many studies? Perhaps a little more background on the development of this program would be helpful. How long has it been going on? Has it been successful? Will it continue?

Response: In lines 72-73 we are defining PPIE in a broad way, regardless of the chronic condition using the most recent definition of the UK National Institute for Health and Care Research (NIHR).In the subsequent lines (74-80), we are trying to show to the reader that recent evidence from original and synthesis studies(ref 10, 11 and 12) demonstrate that PPIE in health research has important advantages. From lines 80-83, we point out the current challenges in implementing PPIE.

All this evidence came from different studies and not from a single experience or program. We tried to make that clear. Please see lines 66-73: “A review of reviews showed that incorporating PPIEP into health research and healthcare can be a powerful tool for identifying patient needs and optimising treatment approaches [11]. PPIEP aims to improve patient outcomes and reduce healthcare costs by empowering patients to make informed decisions and promoting open discussions about disease symptoms and comorbidities [12]. Yet, previous experiences in other context or conditions demonstrate that the effectiveness of PPIEP depends on various factors, including the quality of the relationship between researchers, patients, and the public [13], accessibility of information and training opportunities [14], as well as the health limitations and sociodemographic characteristics of the participants [15].”

4. Introduction, p. 4, lines 104 to 105 – what is the goal of this program after having collected this information? In what manner will it continue?

Response: The current paper does not address the next steps. The current paper focuses on raising awareness that patients are interested in PPIEP initiatives related to asthma. But it also points out that we are captivating only patients with certain characteristics. If broader patient involvement is desired in the future, alternative strategies will be needed to reach and engage a more diverse group.

5. Study Design, p.4 – a little more information about the Patient and Public Involvement Engagement and Participation program would be helpful in terms of the scope of the program, target age, entry criteria, exclusion criteria and also the anticipated burden of the study PPIEP is recruiting for and whether that was included in the invitation to particpate. What were potential participants told would be done with their data and the reason for providing their permission?

Response: Thank you for your valuable comment. We would like to clarify that at the time of this study, no formal PPIEP program existed. The research team conducted this work specifically to assess the potential feasibility of establishing such a network in future. We have now included additional details about the specific question posed to participants to assess their interest. Please see page lines 129-133: “At the end of the stage 2 visit, patients answered a question regarding PPIEP interest (‘We intend to constitute a group of people with asthma with a more active role in research. Your participation will be occasional and will contribute to enrich the research projects - from their conception to the final result - with the perspective and needs of people who deal with asthma day by day. Are you interested in collaborating with this group?’).”.

6. Participants, p.4 – What was the target population surveyed? Were pediatric patients purposely excluded? Was there data collected on their prior participation in asthma or any other studies?

Response: Thank you for your comment. We would like to clarify that the assessment of PPIEP interest was conducted among all adult patients with asthma during stage 2 of the Epi-asthma study. As this is a secondary analysis of Epi-asthma—a study primarily designed to assess adult asthma prevalence in Portugal—our findings are i

---

## [Editor Report · Decision Letter 3]

30 May 2025

Dear Dr. Jácome,

Thank you for submitting your manuscript to PLOS ONE. After careful consideration, we feel that it has merit but does not fully meet PLOS ONE’s publication criteria as it currently stands. Therefore, we invite you to submit a revised version of the manuscript that addresses the points raised during the review process.

We look forward to receiving your revised manuscript.

Kind regards,

Manuela Mendonça Figueirêdo Coelho, Ph.D

Academic Editor

PLOS ONE
---

## [Author Response · Author response to Decision Letter 4]

11 Jun 2025

Dear Prof. Manuela Mendonça Figueirêdo Coelho,

Academic Editor of PLOS One,

June 3rd, 2025

Title: Interest in Public Involvement, Engagement and Participation in Research among Patients with Asthma: A Secondary Analysis of the EPI-ASTHMA Study

Manuscript ID: PONE-D-25-04695

Dear Prof. Manuela Mendonça Figueirêdo Coelho,

Thank you for your consideration of our manuscript and for the useful comments provided.

I am looking forward to hearing from you.

Yours sincerely,

Cristina Jácome

Point-by-point response

Associate Editor:

1. Please, I would like to request the authors to carefully proofread the English.

Response: We have now carefully proofread the manuscript for English language and clarity. All changes made in response to your feedback are highlighted in blue within the revised document.

---

## [Editor Report · Decision Letter 4]

28 Aug 2025

Interest in Public Involvement, Engagement and Participation in Research among Patients with Asthma: A Secondary Analysis of the EPI-ASTHMA Study

PONE-D-25-04695R4

Dear Dr. Jácome,

We’re pleased to inform you that your manuscript has been judged scientifically suitable for publication and will be formally accepted for publication once it meets all outstanding technical requirements.

Kind regards,

Jianhong Zhou

Staff Editor

PLOS ONE
---

## [Editor Report · Acceptance letter]

PONE-D-25-04695R4

PLOS ONE

Dear Dr. Jácome,

I'm pleased to inform you that your manuscript has been deemed suitable for publication in PLOS ONE. Congratulations! Your manuscript is now being handed over to our production team.

Kind regards,

on behalf of

Dr. Jianhong Zhou

Staff Editor

PLOS ONE